# Mechanical and Tribological Properties of Co-Electrodeposited Particulate-Reinforced Metal Matrix Composites: A Critical Review with Interfacial Aspects

**DOI:** 10.3390/ma14123181

**Published:** 2021-06-09

**Authors:** Piotr Jenczyk, Hubert Grzywacz, Michał Milczarek, Dariusz M. Jarząbek

**Affiliations:** 1Institute of Fundamental Technological Research, Polish Academy of Sciences, Pawinskiego 5B, 02-106 Warsaw, Poland; pjenczyk@ippt.pan.pl (P.J.); hgrzywa@ippt.pan.pl (H.G.); mmilcz@ippt.pan.pl (M.M.); 2Faculty of Mechatronics, Warsaw University of Technology, Boboli 8, 02-525 Warsaw, Poland

**Keywords:** experimental mechanics, tribology, co-electrodeposited composites

## Abstract

Particulate-reinforced metal matrix composites (PRMMCs) with excellent tribo-mechanical properties are important engineering materials and have attracted constant scientific interest over the years. Among the various fabrication methods used, co-electrodeposition (CED) is valued due to its efficiency, accuracy, and affordability. However, the way this easy-to-perform process is carried out is inconsistent, with researchers using different methods for volume fraction measurement and tribo-mechanical testing, as well as failing to carry out proper interface characterization. The main contribution of this work lies in its determination of the gaps in the tribo-mechanical research of CED PRMMCs. For mechanical properties, hardness is described with respect to measurement methods, models, and experiments concerning CED PRMMCs. The tribology of such composites is described, taking into account the reinforcement volume fraction, size, and composite fabrication route (direct/pulsed current). Interfacial aspects are discussed using experimental direct strength measurements. Each part includes a critical overview, and future prospects are anticipated. This review paper provides an overview of the tribo-mechanical parameters of Ni-based co-electrodeposited particulate-reinforced metal matrix composite coatings with an interfacial viewpoint and a focus on hardness, wear, and friction behavior.

## 1. Introduction

Composites are materials that combine two or more phases: a continuous one, denoted as a matrix, and an embedded, discontinuous reinforcement. Such structures are preferably used to improve on certain properties of single constituents. For example, one can tailor composites’ tribo-mechanical and physical properties to meet many different and sophisticated application requirements. Furthermore, due to the fact that they can be produced as coatings, it is possible to significantly reduce the cost of application. It should be noted that the technique of surface coating allows the use of cheap, low-quality substrate while still providing a surface with a suitable performance, especially in terms of tribo-mechanical properties. In composite coatings, a matrix can be built from a polymer, ceramic, or metal. Metal matrices are used when elevated temperatures are expected (and thermo-plastic polymers could not suffice) or when ductility is required (and brittle ceramic could not suffice). Ceramic reinforcements are often used, with the most common being Al_2_O_3_ and SiC. Particulate-reinforced metal matrix composites (PRMMCs) are of significant interest due to their lower cost and easier route of processing than fiber-reinforced MMCs. Among the many fabrication techniques used for MMCs [1], co-electrodeposition (CED) provides wide tailoring range yet remains affordable, with repeatable results applicable for complex geometries [2]. Co-electrodeposition means the electrodeposition of a metal coating on a substrate while embedding a reinforcement. Basic types of structures that can be obtained with CED are single-metal, alloy, or multilayer deposits with (nano-)particles, (nano-)wires, or nano-tubes. The deposition can be conducted under direct current (DC), pulsed direct/reverse current PDC/PRC), potentiostatic mode (P), or pulsed potentiostatic mode (PP). Furthermore, there is increasing interest in hybrid composites which are fabricated simultaneously with more than one type of reinforcement [3,4].

Although many reviews concerning different details of MMCs have been published—including reviews on the types of reinforcement used in MMCs [5], Ni-based nanocomposites [6], Ni–P composites [7], electrodeposited Ni-Co/ceramic composites [8], the tribology of electrodeposited Ni-based coatings [9], particle-reinforced Al-matrix composites [10], fiber-reinforced MMCs [11], the mechanical and wear properties of Al-based MMCs [12], the interfacial bonding strength of PR MMCs [13], and interfacial aspects of MMCs [14]—these reviews usually focus on the technology of these materials. Therefore, in this review the fabrication details are not discussed.

The aim of this review is to highlight the gaps in the tribo-mechanical investigations of CED PRMMCs, evaluate the existing methodological approaches and unique insights, and develop a framework to help with further work. Here, we confine this review mainly to Ni-based matrix composites reinforced with particles, mainly SiC; their hardness; and their wear and friction behavior from an interfacial viewpoint due the fact that this is the most popular system used for tribological applications.

## 2. Mechanical Properties: Micro and Nano Hardness

### 2.1. Hardness Measurement Methods

Hardness can be briefly described as a measure of how hard a material is. Although everyone intuitively understands the difference between “hard” and “soft”, hardness can be easily mistaken for some other properties of materials, such as stiffness and yield strength. Therefore, to be more precise, hardness is a measure of how resistant a material is to local plastic deformation. Unfortunately, there is no one specific definition of hardness and no single numerical value that can be linked to a specific material. The definitions of hardness used in modern literature are directly related to the experimental procedure used for its determination. Therefore, hardness cannot be discussed without a description of experimental procedures. Various methods have been invented for the characterization of engineering materials such as metals, ceramics, and composites. Most of these are based on indentation—pushing a small object into a material to induce plastic deformation.

Procedures for macro-scale experiments are standardized and should be performed according to standards. A summary of the most common procedures with references to international standards for further reading is presented in Table 1. Usually, these tests are used for the characterization of bulk samples and use high forces, well above 1 kgf. Due to this, they are not useful for experiments on thin electrodeposited layers. For such purposes, microhardness experiments must be conducted. The two most commonly used methods are Vickers and Knoop micro-indentation. These methods are based on macro-scale procedures and use smaller forces (below 1 kgf). They are also standardized, and references for these standards are given in Table 1.

The general procedure for determining hardness is to indent the surface with a known force and then measure the imprint left on the surface. Depending on the method used, this will be the imprint’s diagonal length, diameter, or depth. This value is then used to calculate the indent projected area based on indentation tip geometry. The value of hardness is given as the maximum indentation force divided by the indent projected area, hence hardness is given in units of pressure. A general formula for indentation hardness is:(1)$$H=\frac{P}{A}$$
where *P* is the maximum force of indentation and *A* is the projected area of the indent.

Standardized micro-scale indentation tests are fast and easy to carry out. Easily available indentation testers are already calibrated to perform experiments according to standards. Due to this, micro-hardness tests are commonly used for the characterization of electrochemically deposited layers. Instrumented indentation is an expanded version of standard indentation. The difference is that, in instrumented indentation, load and the displacement of the indenter are constantly measured and controlled. The result of such indentation is not the only imprint on the surface but also the load–displacement curve, which can be used for further analysis. Although such an approach can be used in every scale of experiment, it is not common for macro-scale experiments. The most widespread method of data analysis was proposed by Oliver and Pharr [15]. It allows the determination of hardness without the measurement of imprint—the depth of indentation is calculated from the shape of the curve. Moreover, it is used to calculate the modulus of elasticity of the material.

### 2.2. Challenges in Experiments

Although indentation tests are easy to perform, there are some general rules to follow and some challenges related to composites to face. First of all, the spacing of indents from each other and the edge of the sample should be wide enough to mitigate the influence of induced stress and deformation to nearby indents. This spacing should be at least 10 times the depth of indent for Berkovich nano-indentation [16] and, in the case of macro/micro Vickers indentation, around three times the diagonal length. For the hardness measurement of thin films, the maximum depth of indentation should not exceed 10% of the thickness [17].

The preparation of the sample before experiments is also crucial. Particularly in micro- and nanoscale indentation, the roughness of the layer strongly influences the results. As shown in the literature [18,19], an increase in roughness from Ra = 0.1 µm to Ra = 1.5 µm can decrease the measured hardness by more than 10%. This effect can be mitigated by the modification of the formulas used for the calculation of hardness. Unfortunately, it is not a standard procedure. This is particularly important in the case of electrodeposited composites. These have usually a high roughness, as particles are incorporated in the top of the layer and covered with metal. The deposition of thicker layers also usually leads to an increase in roughness. Roughness can be lowered by mechanical polishing but, on the other hand, this can induce stress in the layer and subsequently influence the results. Moreover, mechanical polishing can remove weakly bonded particles and change the volume fraction of the reinforcement.

The aforementioned effects will strongly depend on the size of the reinforcement particles used—nanoparticles will minimally influence roughness, while bigger particles will have a stronger effect. The size of the particles also influences the internal structure of the material. This is arguably the hardest factor to take into account, as it cannot be seen with standard instruments. Particles can be counted on the surface or a cross-section of the sample, but this gives no certainty about the homogeneity and particle distribution just below the surface. With small indentations, the agglomeration of particles can drastically change the measured hardness. Mussert et al. [20] give an example of an extreme case when particles are a similar size to the indentation imprint. The change in material being indented during a single experiment is clearly visible as a change in slope in the indentation curve. Particles embedded just below the surface of the specimen will similarly influence the measurement but will be undetectable using the standard micro indentation procedure.

Indentation, particularly instrumented indentation, can also be used in more sophisticated experiments to determine other properties: for example, residual stresses [21], the fracture resistance of interfaces [22], or the adhesion of hard coatings [23,24]. Although interesting, these methods are not commonly used for the characterization of electrodeposited composites.

### 2.3. Models

Hardness is not a strictly defined parameter and is a complex result of the behavior of the internal structure of the material and its interactions with the indenter. Due to this, it is difficult to predict theoretically and calculate it. As composites are mixtures of two different materials, one of the simplest models that can be used is the rule of mixtures (ROM). Two models are described in the literature [25,26]—iso-stress and iso-strain—which can subsequently be used to calculate the lower and upper bonds of such an estimation [25]. Formulas for these models are given below:(2)$$H_{up}=f_{r}H_{r}+f_{m}H_{m},$$
(3)$$H_{low}={\left(\frac{f_{r}}{H_{r}}+\frac{f_{m}}{H_{m}}\right)}^{-1},$$
where *H* is hardness and *f* is the volume fraction, while subscripts denote specific cases: *r* for reinforcement, *m* for matrix, *up* for upper bound, and *low* for lower bound.

Although ROM is easy and convenient, it is too simple in most cases for MMC for the prediction of the actually fabricated sample. Matrices are usually not homogenous and their internal structure strongly influences hardness. Porosity, grain size, dislocations, and internal stresses all affect the mechanical properties of a material. Their influence is most prominent in the case of nano- and microscale experiments, so they are crucial in the case of co-electrodeposited layers. All of these variables are influenced by the deposition process and additives in the bath. The most prominent of these is the effect of grain boundary strengthening, commonly known as the Hall–Petch effect [27]. This is a well-studied phenomenon, especially in the case of electrodeposited nickel.

The most popular numerical simulation method used is the finite element method (FEM). FEM analysis allows the analysis of deformation (both elastic and plastic) and stress distribution in matrices and particles. Two approaches can be described. In the first, an extremely simplified model is created with only one particle embedded in the matrix [28,29]. The second approach is to create a model with numerous particles, either randomly distributed [30,31] or based on the recreated structure of the physical specimen [29,31]. Another simulation method that gained a lot of popularity in recent years, thanks to the growing computational power of computers, is molecular dynamics simulations. They simulate the behavior of individual atoms. In the case of indentation, they are particularly interesting for their ability to simulate the formation of dislocations [32,33]. The biggest drawback is that they are currently capable of simulating only extremely small volumes of material.

### 2.4. Experiments

The measurement of hardness is one of the easiest experiments to perform; consequently, hardness is one of the most commonly described mechanical properties of PR MMCs in the literature. Nonetheless, the comparison of values from various papers is not straightforward. As stated previously, the hardness measured depends on the method used, thus the direct comparison of results from different works is possible only if the same indenter type is used. Moreover, researchers have focused on different types of matrix material (pure nickel or nickel alloy), reinforcement material (SiC, Al_2_O_3_, Si_3_N_4_, etc.), and reinforcement particle size (RPS, from 5 nm to 20 µm), therefore the number of results where direct comparison is possible is limited.

#### 2.4.1. Nickel Matrix: Volume Fraction of Reinforcement

In the first part of a summary of experimental work, we will focus on composites with a pure nickel matrix and a reinforcement consisting of silicon carbide and alumina particles. In addition, we selected only papers with a defined volume fraction of reinforcement. This will allow us to perform a comparison with the ROM model. Vickers micro indentation was used in all the experiments cited in this chapter.

Pinate et al. [34] used Watt’s bath without any additives, deposited using a direct current density of 4 A/dm^2^, and used particles of various sizes (50–500 nm). Additionally, particles were either used as-produced or were surface-treated with HNO_3_. Although the deposition parameters used were the same, the layers had various volume fractions of particles. In general, larger particles and surface treatments led to increased concentrations of SiC, which increased the hardness of the samples. The authors attributed the increase in hardness mainly to the Hall–Petch effect and particle strengthening. They supported these claims using EBSD (electron backscatter diffraction) images of cross-sections of samples, which clearly show the crystal orientation and the size of the individual grains. Microhardness plotted on grain size shows a linear relationship, which is typical for the Hall–Petch effect.

Zhou et al. [35] developed a novel apparatus for co-electrodeposition and tested the composites created with it. Additives (mainly cetyltrimethylammonium bromide) were present in the plating solution. The authors used micro-meter-sized particles and also a relatively high load (500 g) for indentation. The authors reported a higher hardness of pure nickel than that commonly found in papers: 414 HV in comparison to 280 HV. This might be the result of a much higher indentation load. Only one sample with an 18% volume fraction of reinforcement was fabricated; it had a hardness of 626 HV.

The papers of Kiliç [36] and Gül [37] will be analyzed simultaneously, as they employed the same methodology to investigate the influence of different parameters on the mechanical properties of the composite. They used a nickel sulfate bath with SLS (sodium lauryl sulfate) and CTAB (cetyltrimethylammonium bromide) as additives. SiC particles were in the size range of 100–1000 nm. In the first paper [36], the CTAB concentration was varied between 0 and 400 mg/L, while the particle concentration in the electrolyte was 20 g/L. In the second paper [37], the CTAB concentration was constant at 300 mg/L and the particle concentration in the electrolyte was varied between 5 and 30 g/L. All the other parameters of deposition were the same in both cases. The volume fraction of reinforcement increased with the CTAB concentration but not linearly. The highest hardness of the coating was observed in the sample with the second-to-highest CTAB concentration and volume fraction of reinforcements. The authors explained this result by calculating lattice distortion and showing that the highest CTAB concentration decreased the lattice distortion, which is a strengthening mechanism. A higher particle concentration in the electrolyte increased the volume fraction of the reinforcement, with a maximum of 20 g/L. A higher content of particles in the solution slightly decreased the volume fraction. The hardness of deposits changed in the same way. Lattice distortion for these samples showed a weak correlation with the hardness results. Contrary to the previous conclusion, lattice distortion played a lesser role in hardness than dispersion strengthening did.

In another paper, Gül et al. [38] investigated a composite with Al_2_O_3_ particles with a mean size of 80 nm. A Watt’s bath with the addition of SLS and hexadecylpyridinium bromide (HPB) was used. The HPB, particle concentrations in the bath, and also current density were varied. The addition of HPB initially decreased the volume fraction of the reinforcement but higher concentrations reversed it; at the same time, it increased hardness seemingly independent of the volume fraction. The particle content increased both the volume fraction and hardness. Current density had a limited influence on the volume fraction but linearly increased the hardness. The authors also investigated the grain size and lattice distortion in deposits. Both parameters had similar (but reversed) trends with a change in plating parameters.

The comparison of pulsed current (PC) and direct current (DC) deposition methods might show the benefits of either method. Such a comparison is possible thanks to another work carried out by Gül et al. [39], in which the authors used the same bath, additives, and other parameters as in a previous work [38] but changed from DC to 50 Hz PC with a 50% duty cycle. The change to PC increased the volume fraction of the particles in the composite by as much as 40% for higher current densities but increased the hardness by around 5%. On the other hand, DC mode led to grain size that was smaller by around 5% and a slightly higher lattice distortion.

As a summary, we present a graph with all the aforementioned results plotted on a single graph: Figure 1a. The results are also compared with ROM models for both SiC-reinforced and Al_2_O_3_-reinforced composites. For the ROM model, the hardness of nickel was 280 HV, the hardness of SiC was 2563 HV, and the hardness of Al_2_O_3_ was 1600 HV. The graph shows that the ROM model is too simple to accurately predict the hardness of co-electrodeposited composites. The results for SiC-reinforced composites are close to the upper boundary calculated from ROM. Some results are above and others fall below the calculated line. All the results of Gül et al. [38,39] with Al_2_O_3_ as a reinforcement are well above the upper bound.

#### 2.4.2. Pure Nickel Matrix

In this chapter, we gathered the results from composites with a pure nickel matrix that were impossible to directly compare with ROM models. These were specifically from papers that did not provide the volume fraction of reinforcement or that measured hardness using a method other than Vickers. As the variation in the processes used is enormous, only the most important details, conclusions, and remarks about each paper will be presented below. Additionally, we gathered the results in a graph, on which we plotted the range of reported hardness from minimum to maximum. We intend to provide just an overview of the hardness from various papers. Without a single reference value, such as volume fraction before, it is hard to provide a meaningful graph. The number of variables is too high to show them all simultaneously. Vickers hardness is shown in Figure 1b, while Knoop hardness is shown in Figure 1c.

Ha et al. [40] used a sulfamate bath and SiC particles with an average diameter of 50 nm. This publication is focused on the corrosion resistance of coatings. Due to this, few details on the hardness test are provided in the paper, but optical microscope images of indents are shown. In the images, it is evident that the indentation was performed in not ideal conditions. Some indents were placed too close to the substrate, which increases the uncertainty of measurement.

Although nickel sulfate (Watt’s bath) and nickel sulfamate solutions are most commonly used for electrodeposition, other chemicals can be proposed. One of such example is given in the article of Li et al. [41]. In this case, a solution of choline chloride, ethylene glycol, and nickel chloride was prepared as a eutectic mixture and used for electrodeposition. Both nano-sized (40 nm) and micro-sized (300 nm) SiC particles were added as a reinforcement. The concentration of particles in the electrolyte, the stirring rate, and the current density were varied to find the optimal parameters and maximize the SiC weight fraction in the deposit. The coating with nanoparticles had a hardness of 716 HV and the coating with micro-particles had a hardness of 895 HV, both values being much higher than those in any other paper cited in this review. No specific explanation for these high values is given, apart from a general mention of particle strengthening, dispersal strengthening, and grain refining. Therefore, this exceptional hardness was attributed to the plating solution used. Not a single measurement of the grain size, lattice distortion, or quality of the matrix–particle interface was provided to support the hypothesis. It might be inferred that the plating solution promoted a beneficial mixture of the abovementioned properties.

Silicon carbide is one of the hardest materials known—hence its popularity as a reinforcement in hard coatings—but other materials have also been tested. As an example, in the work of Medelien et al. [42], B_4_C was compared with SiC. SiC-reinforced material had a much higher hardness as well as corrosion resistance. Additionally, it is worth noting that micro-particles were used, but the obtained weight fraction of the reinforcement was in the range of 2–5%. Another paper describing coatings with a low weight percent of SiC is that by Vaezi et al. [43]. In this case, the weight fraction of reinforcement were also in the range of 2–5%, but the particles had a diameter of around 50 nm. The reported hardness of coatings was in the range of 540 to 720 HV. Other experiments with the same mass fraction of reinforcement were conducted by Zhou et al. [44]. In this case, a composite containing 6 wt.% nano-SiC was produced from a standard Watt bath. The hardness of such a coating was reported as 550 HV. This shows that the weight fraction was not favorable for the comparison of differently fabricated materials and that a mechanism not dependent on the mass fraction is responsible for the difference in hardness in this case. It is most likely that smaller particles were better dispersed in the matrix and caused stronger dispersion strengthening.

Shathishkumar and Jegan [45] conducted detailed experiments on the influence of the electrical parameters of electrodeposition using a sulfate bath with additives. They used multi-walled carbon nanotubes and SiC as a reinforcement. During their pulsed reverse electrodeposition, they varied the higher cathodic current density (CCD), lower anodic current density (ACD), and anodic current time (ACT). Unfortunately, the authors did not provide any data on either the volume or weight fraction of the reinforcement in the prepared specimens. Due to this, it is hard to draw conclusions on the mechanism that led to a higher hardness.

Indentations conducted with the Knoop method form another distinct subgroup of results. Although the material used might be the same, the hardness measured with another indenter might be different due to the different interactions between them. Aruna et al. [46] explored the influence of sodium hexametaphosphate (SHMP) surfactant and the probe sonication of a particle solution. Both modifications of the process led to higher hardness values. The Knoop hardness test is particularly suitable for the determination of hardness along the thickness of a layer. Thin imprints can be positioned on different locations of a cross-section for the analysis of the depth-dependence of hardness. Kim et al. [47] created a system with two layers of composite with different volume fractions of reinforcement. A layer with 5.4 vol.% had a hardness of around 350 HK, while a layer with 24.6 vol.% exhibited a hardness of around 700 HK. Guo et al. [48] also created a bilayer system. Its hardness changed from 800 to 650 HK from the top to the bottom layer.

#### 2.4.3. Nickel Alloy Matrix

Thanks to the easy electrodeposition of nickel alloys from a single solution, composites with a nickel alloy matrix are also very common. The most popular alloying materials include phosphorus, cobalt, molybdenum, and tungsten. Again, all the cited results have been plotted on a single graph (see Figure 2a, on which we plotted the range of reported hardness from the minimum to the maximum). The material of the matrix is noted above the bars and the material (and diameter) of the reinforcement is given on the horizontal axis.

Examples of composites with Ni–P are given in a series of two publications by Ahmadkhaniha et al. [49,50]. The same methodology was used to analyze the influence of heat treatment on the material and then on the tribological properties of coatings. The alloy was deposited from Watt’s bath with the addition of H_3_PO_3_ as a source of phosphorus. The coatings exhibited a wide range of hardness values depending on the heat treatment used. Additionally, nanoindentation was conducted on selected coatings [49], but no difference in hardness was shown between Ni–P and the composite material. The authors explained this observation as that SiC particles were not blocking dislocations nor introducing a change in microstructure. However, another explanation might also be provided: the volume fraction of reinforcement is too small for it to be possible to notice its influence during nanoindentation.

Ni-Co matrix composites are another popular material. The influence of particle size and volume fraction on the hardness of this material is shown in the article by Bakhit et al. [51]. Composites with nanosized particles exhibited higher hardness values than the composite with micro-sized particles. The volume fraction of the reinforcement was, respectively, 8% and 52%. Pereira et al. [52] created multilayered composites with changing proportions of Ni and Co in the thickness of the layer. Nanoparticles of SiC were used as a reinforcement. Then, a detailed analysis of the hardness of the surface and the cross-section of layers was conducted. A wide range of hardness values was reported, depending on the depth of the coating, composition, grain size, and crystallographic orientation of the matrix.

A less popular material is the Ni-Mo alloy. Its tribology and corrosion properties were investigated by Xu et al. [53]. SiC with a diameter of around 40 nm was used. The deposited layers had a high roughness, and due to this hardness was measured on the cross-section. The authors also measured the grain size of the deposit and, based on that attribute, the change in hardness due to the Hall–Petch effect.

The last group we will discuss is composites based on Ni–W. Li et al. [54] used nanoparticles of Si_3_N_4_ as a reinforcement. They conducted a detailed analysis of pulsed-current deposition parameters on the hardness of both Ni–W and composite material. Based on observations of the alloy, the authors concluded that main cause of high hardness in a composite is in fact the Hall–Petch effect rather than particle strengthening. Cardinal et al. [55] also used PC deposition but obtained layers with a much lower hardness. Although the reinforcement was different—MoS_2_ with a diameter of 3 µm—this was not the reason for the difference. This difference was caused by a high porosity of the layers obtained. The porosity was so high that these layers had a sponge-like structure, which caused the indentation results to be incomparable with others.

#### 2.4.4. Instrumented Indentation

The results of the instrumented indentation are gathered in a single chapter, even though the materials and procedures used are different in each case. Again, we gathered the results from papers in a graph (Figure 2b), on which we plotted the range of reported hardness from the minimum to maximum. The material of the matrix is noted above bars and the material (and diameter) of the reinforcement is given on the horizontal axis.

Tsongas et al. [56] prepared Ni–P coatings using PC deposition with SiC particles as a reinforcement. Additionally, the composites were heat-treated (HT) at 400 °C for 1 h. This treatment increased the hardness of both the Ni–P alloy and the composite. The elastic modulus determined with the Oliver and Pharr method is also presented. The use of instrumented indentation allows noticing the different behavior of hardness and the elastic modulus. Samples had a hardness in ascending order of Ni–P > Ni–P/SiC > Ni–P HT > Ni–P/SiC HT, while elastic modulus changed from Ni–P > Ni–P HT > Ni–P/SiC > Ni–P/SiC HT. This shows that a different mechanism influences hardness and elastic modulus, but this is not explored in this paper. Additionally, the authors used the FEM to predict stress–strain curves based on the indentation results.

Wasekar et al. [57,58] worked on composites with Ni–W alloy as a matrix and 350 nm SiC particles were created with PC electrodeposition. Two papers concerning a similar material were prepared; in the first [57], electrical parameters were investigated, and in the second [58] the content of particles in the electrolyte was varied. In both cases, the hardness and elastic modulus were increasing in up to 5% of the volume fraction of the reinforcement, at which point the trend was reversed. This is explained by the authors through the inverse Hall–Petch effect.

Jenczyk et al. [59] produced coatings with large particles with a mean diameter of 17 µm. The novel idea in this work was to coat SiC particles with a protective layer of nickel using physical vapor deposition (PVD). Contrary to previously cited works, they used an instrumented Vickers indenter with a maximum force of 10 N, leaving imprints 12–20 µm deep with a diagonal of around 40 µm. The authors note that the results are burdened with a large uncertainty, as the imprints were too small for such big particles, and at the same time, larger imprints could not be carried out because of the small thickness of the layers.

### 2.5. Critical Overview

Although hardness is a commonly investigated property, it is often not analyzed carefully enough. In most cases, it is analyzed as an additional, easy-to-perform measurement with the use of standard micro-indentation devices. Instrumented indentation is rarely used, even though it provides far more information about the mechanical properties of a material. The comparison of various results is complicated, as there is no standard way to determine the fraction of reinforcement in a composite. Some researchers have calculated the mass fraction from X-ray diffraction (XRD) analysis; others use volume fraction based on microscope images, and others do not even specify this. The analysis of other parameters such as lattice distortion and grain size is also rare. Without knowing more parameters for the structure of a composite, it is nearly impossible to determine which mechanism is responsible for the change in hardness.

Barely any of the cited papers provided images of imprints. We find this to be a crucial deficiency, as such images provide important details about indentation. First of all, they are a proof that indentation was conducted in a proper way—with proper spacing, etc. Moreover, they provide additional data that could be used to improve the data analysis. Atomic force microscopy (AFM) images show the pile-up or sink-in behavior of a material. The measurement of this parameter would allow the calculation of hardness with a higher precision. Even a simple picture from an optical microscope (OM) can show the distribution of particles under and in proximity to the indent.

## 3. Tribological Properties: Wear Resistance under Dry Sliding Conditions

### 3.1. Mechanisms of Wear Resistance of Particle-Reinforced Composites

Wear is the process of removing material from surfaces that are in motion relative to each other [60]. Particle-reinforced composite coatings, such as advanced engineering materials, are used in many applications where a high wear resistance is required. Examples of applications include electric brushes, cylinder liners, artificial joints, and helicopter blades. Indeed, compared to conventional materials, the introduction of particles into a metal matrix significantly increases the wear resistance. In addition, the wear resistance can be strongly modified by such changes in the microstructure as modification of the morphology, the selection of an appropriate volume fraction, the size and type of reinforcing particles used, and the nature of the connection between the reinforcement and the matrix. In order to obtain optimal wear properties without the deterioration of beneficial matrix properties, it is very important to accurately predict the wear of composites and thus to understand the wear mechanisms involved. Although a number of studies have been carried out on metal matrix particle-reinforced systems, the wear mechanisms in these composites are far from being fully understood. Usually, three wear mechanisms—abrasion, oxidation, and adhesion—are observed in these systems depending on the constituent materials, applied load, and sliding distance [61].

Abrasive wear occurs when a harder material rubs against a softer material. In the case of PRMMCs, the wear is almost entirely caused by hard reinforcement particles trapped between the rubbing surfaces. These particles are embedded into the metal matrix only at the beginning of the wear process. Over time, the particles are released into the tribosystem and unfavorable three-body wear occurs. The reinforcement is removed due to failure at the matrix/reinforcement interface or in the reinforcement. The interfacial bonding between constituent materials may be weak due to chemical incompatibility; mismatch in thermal expansion; or the elastic properties—e.g., stiffness—at the interface [62]. On a microscale, a single released particle can wear a surface in three different wear modes: ploughing, cutting, and cracking.

If the ploughing occurs then the material is shifted to the sides of the wear grooves. In this case, the material is not removed from any one of the matching surfaces. On the other hand, in the case of cutting the material is removed in the form of chips in front of a particle. Finally, cracking means that cracks are formed in the subsurface regions surrounding the wear grooves [60].

Furthermore, adhesive wear is the result of micro-joints caused by welding between opposing asperities on the rubbing surfaces of the counterbodies. The load applied to the contacting asperities is so great that the micro-joint junctions deform and adhere to each other. The movement of rubbing counterbodies results in the rupture of the micro-joints. Welded roughness causes cracks in non-deformed regions.

Wear can be accelerated by the oxidation (corrosion) of rubbing surfaces. Increased temperatures and the removal of protective oxide layers from the surface during friction promote the oxidation process. Friction ensures that the oxide film is continuously removed before the formation of new oxide occurs. The hard oxide particles removed from the surface and trapped by the sliding/rolling surfaces additionally increase the wear rate by the three-body abrasive wear mechanism.

### 3.2. Models

In order to model and precisely predict wear, especially PRMMC wear, one should take many parameters into account—i.e., the mechanical properties of both counter bodies, interfacial bonding strength, oxidation resistance, adhesion, etc. Hence, PRMMC analytical models are usually highly simplified. Usually, one takes into consideration only one wear mechanism that is dominant in the system.

This approach is represented in many papers [62,63,64]. The simplest models of abrasive wear of PRMMCs are based on two equations. The first one, called the inverse rule of mixtures, was introduced for two-phase composites by Khruschov and Babichev [65]. In this case, the wear rate of a composite (WC) can be determined as follows:(4)$$\frac{1}{W_{C}}=\frac{V_{M}}{W_{M}}+\frac{V_{R}}{W_{R}}$$
where *W* and *V* are, respectively, the wear rates and volume fractions of the matrix (designated by subscript *M*) and reinforcement (designated by *R*). This equation is based on the assumption that the components of the composite wear at an equal rate. Hence, it predicts that the abrasive wear resistance is linearly additive and that the wear resistance of the composite is simply the sum of the products of wear resistance and volume fraction for each component. Another example of this approach, proposed firstly by Zum-Gahr [66], assumes that the proportion of each component is linearly proportional to its volume fraction in the composite:(5)$$W_{C}=V_{M}W_{M}+V_{R}W_{R}$$

In this model, the intensity of abrasive wear decreases linearly with the growth of the volume fraction of the reinforcement. Unfortunately, these extremely simplified models are not confirmed by the experimental results due to their non-physical nature. In Figure 3, the models and example results of wear rate are indicated. It is clear that these models do not reflect the reality of the situation. Both rely on the assumption that all the components in the composite wear in the same way as they would in a bulk material. This affects the volume rate of volume fractions and the wear rate of the components. This approach does not take into consideration other important factors, such as the interfacial properties between the distinctive phases, the relative sizes, and the fracture toughness of these phases. However, in many papers it has been demonstrated that these factors have a significant influence on abrasion in composites—i.e., due to the fact that the particles released to the tribosystem cause more intensive, three-body wear.

An analytical model which takes into account the aforementioned factors was proposed by Lee et al. They have developed an abrasive wear model based on three primary wear mechanisms: plowing, cracking at the interface or in the reinforcement, and particle removal [62]. In this model, the interfacial properties and geometrical and mechanical properties of the reinforcement are considered by introducing a factor related to the fracture toughness of the matrix/reinforcement interface and the reinforcement, as well as the relative size of the reinforcement relative to the abrasive grains. It is then possible to predict the negative reinforcement effect and dependence of the interface and reinforcement size on the wear rate. Due to the progress in computer science, the analytical models are replaced by numerical simulations—i.e., with the use of the finite element method [67,68].

Another approach is to mix experimental results, statistical analyses, and extrapolation in order to predict the wear rate. In this case, the Taguchi method is often used [69,70,71]. This is a method which optimizes design parameters through the use of special orthogonal arrays to study all the design factors with a minimum number of experiments.

### 3.3. Experiments

To date, in many experimental works it has been proven that a uniform dispersion of coelectrodeposited particles leads to an improvement in the wear resistance of coatings. In particular, Ni/SiC composites exhibit a high wear resistance, which has allowed their commercialization for the protection of friction parts, combustion engines, and casting molds [72,73]. Similarly to section on friction coefficients, we will discuss the effects of the co-electrodeposition method (DC or PC), volume fraction, and particle size on the wear resistance.

It should be noted that wear is usually determined from the volumetric wear factor *C_W_*, which is defined as follows:(6)$$C_{W}=\frac{V}{FS}$$
where *V* is the volume loss determined—i.e., by profilometry; *F* is the applied load; and S is the total sliding distance. Generally, comparison between different papers is rather difficult and may be burdened with large error due to the fact that the wear is measured with many different setups and in some cases other quantities determined by wear (i.e., mass loss or the wear track depth). Furthermore, it is usually assumed that the wear is described by the Archard law, which states that the wear rate is linearly dependent on the sliding distance and applied load and inversely proportional to the hardness, which for PRMMCs may not always be true.

#### 3.3.1. Direct/Pulse Current Electrodeposition

Gyftou et al. created Ni–SiC coelectrodeposited coatings through the use of DC and PC with different duty cycles [72]. In their work, in order to study wear resistance they used a pin-on-disc measurement, in which the coatings were tested against corundum balls (r = 3 mm). Finally, in this work it was shown that PC electrodeposition results in a significantly higher wear resistance. The volumetric wear factor for microparticles (1 µm) was equal to 6 × 10^−8^ cm^3^/Nm for DC, whereas for PC it was equal to about 30 × 10^−4^ mm^3^/Nm for a 10% duty cycle and about 10 × 10^−4^ mm^3^/Nm for higher values of duty cycle (up to 90%). Similarly, in the case of nanoparticles (20 nm) the wear factor was lower for the PC but significantly influenced by the duty cycle. The lowest value (below 10 × 10^−4^ mm^3^) was observed for a duty cycle equal to 50%. The authors also investigated the wear tracks and concluded that the main wear mechanism was abrasive wear—clearly visible scratches parallel to the sliding direction.

The same effect was observed by Gül et al. for the Ni–Al_2_O_3_ composite [38,39]. In this study, coatings were prepared from a modified Watt-type electrolyte by both DC and PC plating under current densities varying between 1 and 9 A/dm^2^. The coatings were tested against an M50 steel ball (r = 5 mm). In almost all the cases studied in this work, the PC-coated composites exhibited significantly lower wear rates. For example, when the wear rates of the DC- and PC-coated nanocomposites were compared in the case of a 1 A/dm^2^ current density for the sliding speed of 150 mm/s, the wear rate was recorded as 17 × 10^−4^ mm^3^/Nm in the DC-plated material, whereas the wear rate was measured as 2 × 10^−4^ mm^3^/Nm in the PC-produced nanocomposite. Therefore, the wear rate of the PC-deposited coating was found to be approximately eight times lower than that of DC-plated material for a 1 A/dm^2^ current density deposition condition. An exception from this rule was only observed for the sliding speed of 50 mm/s—in this case, the wear rates were similar and, depending on the current density, between 6 and 10 × 10^−4^ mm^3^/Nm. Interestingly, the studies of the wear tracks revealed that the mechanism of wear significantly depended on both the deposition technology (DC or PC) and sliding velocity. For DC for all the studied velocities, a significant particle delamination occurred and therefore abrasive wear dominated. In the case of PC for lower velocities (50 mm/s), a mixture of abrasive and adhesion wear was observed, whereas for higher velocities (150 mm/s) the main mechanism was the abrasive wear, but this was also supported by significant oxidation.

#### 3.3.2. Volume Fraction and Distribution

Firstly, it should be noted that the concentration in deposits is not a linear function of concentration in the electrolyte and depends on many other parameters, such as current density, particle size, and agitation during deposition [74]. One should also take into account that the mechanical properties of composites of the same volume fraction may vary depending on the distribution and size of the dispersed phase [75]. For example, the mechanisms that are probably responsible for the higher wear resistance of PC-deposited coatings are the better reinforcement dispersion and higher adhesion between the reinforcement and the matrix. The latter is discussed in the paragraph about interfacial strength. A better dispersion due to PC deposition has been shown in several papers [54,76,77]. For example, Li et al. showed that the PC deposition technology results in a more uniform dispersion of Si_3_N_4_ particles in a Ni–W matrix, which results in a higher hardness, lower coefficient of friction, and higher wear resistance [54].

Furthermore, the concentration of the particles has a significant influence on the wear resistance [78,79]. However, it should be noted that in general it is not true that the higher the concentration of particles, the greater the wear resistance. For example, Garcia et al., who investigated Ni–SiC coatings, showed that the wear resistance reaches the minimum for a concentration of particles equal to about 5% [75]. For higher concentrations, the wear resistance decreases, and for concentrations higher than about 20% it is worse than in the case of using a pure Ni coating. Similar results have been shown by Jenczyk et al. [59] and Kilic et al. [37]. Jenczyk et al. investigated Ni–SiC coatings observed a significant decrease in wear resistance for concentrations higher than 10%. However, the authors also showed that this may be strongly increased by the improvement of the interfacial bonding strength between particles and the matrix. When they applied a Cu protective layer to SiC particles in order to prevent the creation of a carbon layer on the particles in the electrolyte, then they observed a significant increase in the wear resistance for higher particle concentrations. Kilic et al. studied Ni–nanoSiC coatings deposited from the standard Watt bath with the addition of cetyltrimethylammonium bromide (CTAB) and sodyumdodecyl sulfate. They observed a maximum wear resistance at a 10.05% particle concentration. Additionally, they observed that for higher particle concentrations, the wear mechanism switches from cracking to abrasive wear. Unfortunately, it is not clear if this effect is caused by the application of the surfactant or if it is the result of only the particle concentration. The authors also did not discuss the influence of the surfactant on the interfacial bonding strength.

On the other hand, Shrestha et al. [80] presented results showing that the maximum wear resistance was obtained for 50 vol.% SiC. They studied Ni–SiC coatings obtained by co-electrodeposition from a bath containing cationic surfactants with an azobenzene group (AZTAB). It may be then possible that this particular addition may significantly increase not only the SiC concentration (as shown in the paper) but also the interfacial bonding strength.

Narasimman et al. [81] obtained results showing a middle ground between those presented by Shresta and Jenczyk or Kilic. In their paper, they studied Ni/SiC composites with micro and nanoparticles. The minimum wear was obtained at a 24% particle concentration of nanoparticles and at a 29% particle concentration for microparticles.

There are also papers in which the authors claim that wear resistance increases monotonically with the particle concentration. For example, Wasekar et al. showed such results in their paper [58]. They claim that the wear mechanism can be best explained using the inverse rule of mixtures. However, in light of the previously discussed results, this claim is only true for low particle concentrations. Wasekar et al. studied Ni–W/SiC coatings containing 0–6 vol.% of submicron (350 nm) SiC particles. The wear rate starts at 1.3 × 10^−6^ mm^3^/Nm and reaches 0.3 × 10^−6^ mm^3^/Nm. Furthermore, in the case of Ni–Mo/diamond coatings, Liu et al. reported that the wear resistance increases with the increase in the concentration of diamond particles [82]. They studied concentrations between 0 and 21 vol.% and obtained a wear rate equal to 3.5 × 10^−6^ mm^3^/Nm for a 0% concentration and about 0.25 × 10^−6^ mm^3^/Nm for a 21% concentration.

To sum up, in this paper we have tried to plot the wear resistance vs. particle concentration (Figure 3) obtained in many different papers. This task is not easy, due to the fact that there are many different systems (i.e., different nickel alloys or reinforced particles) and wear parameters used. Hence, we have developed a method to plot a so-called normalized wear rate. We obtain this by dividing the wear parameters presented in a given work by the wear parameter measured for the lowest particle concentration (usually 0 vol.%, but not every work gives such a reference). According to this plot, a reduction in wear is a function of the particle concentration, but its course is significantly different from the rule of mixtures (dashed gray line) or the linear wear model [66] (dotted gray line). For the rule of mixtures, it was assumed that the wear rate of SiC is five times lower than for nickel alloy coatings. The experimental function has the minimum; however, the particle concentration corresponding to that minimum has not yet been determined. Most observations connect the behavior of the wear rate vs. particle concentration function with the interfacial bonding strength [59,83], but this issue needs to be further investigated.

#### 3.3.3. Particle Size

Usually, nanoparticles are particles with a size lower than 100 nm. Hence, in most of the papers cited in this work, microparticles were applied. The particle size is indicated in Figure 3, but due to the significantly different systems used in different works, it is rather difficult to state unequivocally what the influence of the particle size is on this basis. Fortunately, Narasimman et al. studied the difference in wear rate between nanoparticle (~50 nm) and microparticle (~1 µm) deposits [81]. Their deposits differ from each other only in terms of the particles used. In Figure 3, nanoparticles are indicated by unfilled circles. On the basis of this research, one can state that the coatings with nanoparticles exhibit lower wear rates and that the minimum for the wear rate vs. particle concentration function is moved towards higher particle concentrations. This is consistent with the studies performed for other metal matrix particle-reinforced composites—i.e., Sadooghi and Rahmani showed that Mg/nanoSiC composites produced by the friction stir process have a lower wear rate than their Mg/microSiC equivalents [84]. Furthermore, although Garcia et al. have not studied nanoparticles in the way described above, they have studied the influence of the particle size on co-deposited coatings. They applied particles with sizes of 0.3, 0.7, and 5 µm and concluded that the particle size significantly influences the particle concentration in the coatings. Smaller particles are easier to co-deposit. However, from their work it is not clear if smaller particles lead to a higher wear resistance.

### 3.4. Critical Overview

The biggest problem in the investigation of the wear of PRMMC composites is the lack of reliable models. Models should include many parameters and sophisticated assumptions; hence, numerical models may be difficult to carry out even using modern supercomputers. Hence, one should concentrate on the development of analytical models for simple parts of the wear mechanism and then combine them in multiscale numerical/analytical models. This is a difficult task, and it is not yet clear if it is even possible to obtain any reliable results.

Therefore, the experimental approach is still the only one reliable in this field. Unfortunately, the biggest problem in comparing different experimental works in which wear resistance is studied is the multitude of parameters and measurement techniques used. Authors usually do not make enough effort to relate their results to those of other works. A normalized wear rate presented in this paper is an attempt to do that, but it still has some disadvantages—i.e., it is clear that the particle size plays a significant role in the wear process, but this is not taken into account by this simple parameter. Moreover, in the case of particle size one of the more interesting problems which should be investigated is the higher wear rate of nanocomposites. Is this caused by the particle properties (higher mechanical strength), better matrix–particle interface, or something else?

## 4. Tribological Properties: Friction under Dry Sliding Conditions

### 4.1. Rules and Mechanisms of Friction in Metals

Friction is the resistance to movement during slip or rolling that occurs when one body is constantly moving tangentially toward another with which it is in contact. The tangential resistance force which acts in the opposite direction to motion is called the friction force. There are two main types of friction: dry and fluid friction. Dry friction, also known as “Coulomb” friction, describes the tangential component of the contact force that occurs when two dry surfaces move or tend to move relative to each other [60]. The first law of Amonton (1699) states that the coefficient of friction (CoF) is a dimensionless scalar value that represents the ratio of the frictional force *F* between two bodies to the normal force *W* pressing them together:(7)$$\mu =\frac{F}{W}$$
in which *μ* is the symbol of CoF.

The second law of Amonton states that contact between the contacting bodies has no influence on the friction force. In most cases (particularly at the micro and nanoscale), this rule is not true because of the surface properties—e.g., roughness. Friction is modelled by theories of contact mechanics (the most popular models are those of Hertz [85], Johnson–Kendall–Roberts (JKR) [86], and Derjaguin-Muller-Toporov (DMT) [87]; the last two include adhesive forces). At the macroscale, CoF values are less than one (~0.1 for polished surfaces, ~0.4 to ~0.8 for asperity contact). On the other hand, CoF values can be more than one, in particular at the nanoscale [85].

Due to the insufficient correspondence between Coulomb theory [60] and experiments, Bowden and Tabor (1950) proposed a friction model that takes into account adhesive and deformation interactions. When two metals are sliding in contact, a high pressure is created at the points of contact, causing spot welding. The contacts created in this way are successively sheared by the relative surface sliding. In addition to the frictional energy which is necessary to overcome the adhesion developed at the actual contact areas between the surfaces (roughness contacts), energy is also required to plastic deform the contacting surfaces at the microscale during motion. As the roughness of one surface passes through the other by plastic deformation, energy is required for macro-scale deformation (grooving or ploughing). Macro-scale deformation can also occur through particles, a third body, trapped between the sliding surfaces.

In 1964, Bowden and Tabor suggested a model, in which the total frictional force (*F_i_*) is equal to the force needed to shear adhered junctions (*F_a_*) and the force needed to supply the energy of deformation (*F_d_*) [60]:(8)$$F_{i}=F_{a}+F_{d}$$
(9)$$\mu_{i}=\mu_{a}+\mu_{d}$$
where *μ_i_, μ_a_,* and *μ_d_* are the coefficients of friction: fractional, adhesional, and deformational, respectively. The last two CoFs are described in Section 4.2 and Section 4.3.

### 4.2. Adhesional Coefficient of Friction (μ_a_)

The real contact area *A_r_* is the sum of the areas of the contact points. In most cases, under normal load this is a small fraction of the apparent contact area *A_a_*. Adhesive contact is caused by the roughness of the surface. As the two bodies move relative to each other, a lateral force is required to break the aforementioned adhesive forces. The result is cracks at the weakest points, at the joint, or at one of the fasteners. Bowden and Tabor defined the Fa friction force in 1950 (for dry contact) [60]:(10)$$F_{a}=A_{r}\tau_{a}$$
where *τ_a_* is the average shear strength of the dry contact. The coefficient of adhesional friction is:(11)$$\mu_{a}=\frac{A_{r}\tau_{a}}{W}=\frac{\tau_{a}}{p_{r}}$$
where *p_r_* is the mean real pressure. The mechanism of friction is divided into the elastic and plastic regimes. For elastic theory, *μ_a_* is proportional to the formula:(12)$$\mu_{a}\;\sim \;\frac{3.2\tau_{a}}{E^{*}\sqrt{\frac{\sigma_{p}}{R_{p}}}}$$
where *E** is the composite (or effective) elastic modulus and *σ_p_* and *R_p_* are the standard deviations of the summit heights and average summit radius, respectively. For plastic contacts:(13)$$\mu_{a}=\frac{\tau_{a}}{H}$$
where *H* is the hardness of the softer of the contacting materials. For a plastic regime, the adhesional coefficient of friction is independent of the surface roughness, unlike that in elastic contacts [60].

### 4.3. Deformational Coefficient of Friction (μ_d_)

When two surfaces slip, two types of interaction can occur: microscopic interaction (plastic deformation and roughness displacement) and macroscopic interaction, in which the roughness of a harder material either ploughs grooves on the surface of the softer surface by plastic deformation or causes cracks, tears, or fragmentation. In addition, plowing increases the frictional force but can also introduce wear particles, which increase subsequent friction and wear [60].

During the sliding process, the front surface of the asperity is in contact with the softer body. The load-support area *A_l_* is shown in equation:(14)$$W=pA_{l}$$
where *p* is the yield pressure (the yielding of the body is isotropic). The friction force is supported by the grooved area *A_p_*:(15)$$F=pA_{p}$$

The ploughing component of the coefficient of friction is:(16)$$\mu_{d}=\frac{F}{W}=\frac{A_{l}}{A_{p}}$$

In the calculations of model asperities, the pile-up of material has been neglected. However, its contribution in some cases may be significant [60].

### 4.4. Friction of Metal Matrix Composites

Reinforcement in MMCs may improve the mechanical and tribological properties compared to pure matrix material. Al_2_O_3_, SiC, and B_4_C are very common reinforcements in MMCs. On the other hand, usually hard ceramic particles lead to increases in CoF [37,88,89,90]. One method which is used to decrease the CoF of materials is embedding suitable reinforcements known as solid lubricants into the metal matrices to produce composites, such as graphite and MoS_2_. The parameters that can influence the friction behavior of MMCs can be classified into three categories [91]:Material factors;Mechanical factors;Physical factors.

It should be noted that material factors depend on the fabrication route (direct/pulse current, magnetic-assisted electrodeposition, and post-deposition (heat) treatment) and its parameters (current density, time, electrolyte composition, stirring, temperature).

The dry friction mechanism in MMCs is mainly based on the friction mechanism proposed by Bowden and Tabor (see Section 4.2 and Section 4.3). Reinforcement particles inside the metal matrix can prevent the plastic deformation of the matrix. On the other hand, particles torn out of the matrix may become a third body and can increase ploughing and, consequently, friction [91].

In general, the composites with a solid lubricating reinforcement show better tribological properties—e.g., Ni–P/MoS_2_ coating. During the friction process, MoS_2_ particles form a densely packed layer of lubricating coating. This may be the case due to MoS_2_ accumulation in wear crevices. The lubricating film adheres to the surface of the coating; excellent tribological properties, such as low CoF and low wear factor, are observed in this coating [92].

Most often, the CoF is calculated with Equation (7) without distinguishing adhesional and deformational factors. In addition, statistical models can be used to study the material, mechanical, and physical factors of MMCs on the CoF—e.g., the Taguchi method ([93,94]), response surface model (RSM) [95], artificial neural network (ANN) [96,97], grey relational analysis (GRA) [71], and analysis of variance (ANOVA) [98,99].

ANOVA can show the relative influence of the investigated parameters on the CoF. For instance, Al/SiC/MoS_2_ hybrid MMCs were investigated in high-temperature environmental conditions. The analysis of variance showed the influence of the parameters of temperature (39.85%), weight percentage of SiC (34.60%), sliding velocity (14.57%), particle size of SiC (4.80%), load (3.92%), and sliding distance (1.55%) on the CoF. In addition, a mathematical model for CoF is:(17)$$CoF=0.1958+4.62\cdot 10^{-4}\cdot P-3.9\cdot 10^{-3}\cdot W+3.16\cdot 10^{-4}\cdot L+1.24\cdot 10^{-2}\cdot V\;+4.21\cdot 10^{-4}\cdot T+8.33\cdot 10^{-7}\cdot S$$
where *P* is particle size, *W* is weight, *L* is load, *V* is sliding velocity, *T* is temperature, and *S* is sliding distance [99].

F. Rana et al. investigated the influence of material factors (*M1* and *M2*: weight percent SiC and particle size, respectively) and processing factors (*P2* and *P3*: mixing temperature and speed, respectively, in addition to the mixing time and feed rate of particles) on the CoF of Al-1.5% Mg/SiC particulate MMCs. The regression analysis model yielded an equation of the form:(18)$$CoF=1.225-7.8\cdot 10^{-2}\cdot M1+10^{-3}\cdot M2+1.5\cdot 10^{-4}\cdot P2-1.67\cdot 10^{-5}\cdot P3.$$

The equation confirmed the fact that the CoF depends on the particle size and concentration of SiC particles. A better dispersion of the particles also decreases the CoF. Since processing parameters such as mixing temperature and mixing speed affect the dispersion of particles, they will influence the frictional properties [100].

Statistical models can also be used to optimize current parameters (such as the frequency, duty cycle, and current density), to obtain MMCs with tailored CoFs. For instance, the optimal electroplating parameters were obtained through grey relational analysis (GRA). The optimal Ni/SiC nanocomposite coating parameters were predicted to be a frequency of 10 Hz, duty cycle of, 10% and current density of 0.2 Acm^−2^, which is in agreement with experimental confirmations [71].

### 4.5. Experimental Investigations

In this section, we focused on the friction properties of CED PRMMCs. The main focus is Ni-based matrices with SiC, Al_2_O_3_, or MoS_2_ reinforcements. Sample preparation parameters (i.e., direct/pulse current electrodeposition, reinforcement particle size, volume fraction, heat treatment) and their influence on the CoF are discussed.

The most common methods for carrying out friction and wear tests are ball-on-disc and pin-on-disc tribometers. In that papers, MMC and steel were friction pair, with a steel ball diameter ranging from 3 to 15 mm. For the measurement of the wear and friction tests, balls made of bearing steel with the following names were used: SUJ2, EN31, GCr15, SAE 52100, AISI 52100, AISI 8620. Balls made of corundum, Al_2_O_3_, and Cr were used in several articles. Diamond spheres were characterized by relatively small diameters equal to 100 μm. In pin-on-disc tribometers, the most common disc material was WC-Co and EN 31 hardened steel. In a large minority of the articles, ring-on-disc-pair tribometers were used. The most common friction pair with MMC was JIS SKD11 steel. The CoF diagrams include friction measurements at room temperature for MMCs without annealing.

#### 4.5.1. Direct/Pulse Current (DC/PC) Electrodeposition and Methods Modification

For MMC electrodeposition, two types of power supply can be used: DC ([36,37,38,48,49,50,59,80,90,101,102,103,104,105,106,107,108,109,110]) and PC ([41,55,58,78,111,112,113,114,115,116]). In addition, the effect of DC or PC deposition was investigated in several papers ([38,39,54,72,117]). The power supply parameters, such as current density, frequency, and duty cycle, can strongly influence MMC mechanical properties. In this section, we focused on the evaluation of CoF.

Two types of electrodeposition based on DC and PC plating were used to prepare Ni–W/Si_3_N_4_ composites. H. Li et al. prepared samples of Ni–W, Ni–W/S_i3_N_4_ (DC), and Ni–W/Si_3_N_4_ (PC). The reinforcement particle size was 20 nm. A 3–4-times lower CoF was found for composites compared to a pure Ni–W layer. In addition, the Ni–W/Si_3_N_4_ (PC) composite had a lower CoF compared to Ni–W/Si_3_N_4_ (DC). The authors have provided a tentative explanation: the improvement in wear resistance was observed for the composite coating prepared by the PC method, which was characterized by smaller grain sizes and a higher density, as compared to DC electrodeposition [54].

The opposite behavior was observed by P. Gyftou et al. for Ni/SiC composites. For comparison, Ni/SiC (DC) and Ni/SiC (PC, duty cycle: 10, 30, 50, 70, 90%) samples were prepared. The CoF of Ni/SiC (DC) was approximately 20% lower than the CoF of Ni/SiC (PC, 1 μm SiC). On the other hand, for layers with 20 nm SiC particles, the CoF was slightly lower for the PC sample [72].

C. Sun et al. used magnetic-assisted pulse electrodeposition to prepare Ni/SiC nanocoatings. They investigated a pure Ni layer (duty cycle 20%, magnetic intensity 0.4 T) and three Ni/SiC composites (duty cycle 20%, magnetic intensity: 0.2, 0.4, 0.6 T). The pure Ni coating exhibited the highest CoF and the mean CoF was about 0.85. The Ni/SiC MMCs deposited at 0.6 T had the smallest CoF (~0.44). Smaller CoFs were observed for denser and smoother coatings [112].

H. Gül et al. prepared Ni/Al_2_O_3_ samples using DC plating (current density: 1, 3, 6, 9 A/dm^2^) with a modified Watt-type electrolyte. It was noticed that the lowest current density resulted in the lowest Al_2_O_3_ volume percentage in the coatings (3.98%). For higher current densities, the reinforcement volume percentage was stable (8.46–8.81%). With an increase in the current density, the CoF decreased (from 0.7 to 0.2) [38]. In subsequent work, H. Gül et al. compared Ni/Al_2_O_3_ CoFs again. Four DC-plated Ni/Al_2_O_3_ samples (current density: 1, 3, 6, 9 A/dm^2^) and four PC-plated Ni/Al_2_O_3_ samples (the same current densities, duty cycle 50%) were prepared. In addition, they investigated the influence of sliding speed (50, 100, 150 mm/s) on CoF. With the increase in the sliding speed for DC and PC coatings, the CoF was reduced. For 100 and 150 mm/s, the CoF values were similar with both the DC and PC electrodeposition methods. On the other hand, for DC composites slight increases in CoF values were found when the current density increased. The CoF values for DC MMCs were relatively high—approximately 0.7 for current densities of 1 A/dm^2^ and 3 A/dm^2^. In addition, with an increase in the current density the CoF values dropped. As a comparison, the CoF values for the PC electrodeposited MMCs were stable (around 0.43–0.52) [39].

Electrophoresis-electrodeposited MMCs were studied in several papers [[106,113]]. Y. Zhang et al. prepared Ni-Co/Al_2_O_3_-MoS_2_ composites with different MoS2 concentrations. With the increase in the MoS_2_ bath concentration, the MoS_2_ volume fraction in the MMCs increased (7.7–15.95%). At the start of the experiment, the MoS_2_ molecules did not act as lubrication, perhaps due to the insufficient MoS2 detachment. However, over time a large number of MoS_2_ particles were pulled out of the matrix. A self-lubricating film was formed between the bodies. They observed decreasing CoF values for Ni–Co/Al_2_O_3_–MoS_2_ coatings. Hard reinforcement particles prevented plastic deformation throughout the friction and wear tests [104]. Z. Jia et al. investigated the influence of pulse/electrophoresis electrodeposition. Ni–Co/Al_2_O_3_ samples were prepared using both methods and their tribology properties were investigated. With the increase in the experimental time, the CoF values of the MMCs prepared using the PC method began to be higher than the CoF values of the MMCs prepared using the electrophoresis–electrodeposition method. The CoF reached about 0.45 for an electrophoresis–electrodeposited sample, whereas for pulsed electrodeposition, the CoF was higher and equal to 0.65. The difference was attributed to the higher hardness of the composite coating prepared by electrophoresis–electrodeposition [113].

#### 4.5.2. Reinforcement Volume Fraction (vol.%)

To obtain different vol.% of reinforcement in composites, several methods or components are modified. The most common method is to change the reinforcement particle concentration in the electrolyte ([37,75,78,109]), the additives in the electrolyte ([36,80,109]), or the electrodeposition current parameters ([38,39,72,113]).

H. Gül et al. investigated the effect of particle concentration in the DC-plated Ni/SiC composites. Four types of particle concentration on electrolyte of 5, 10, 20, and 30 g/L were used. The CoF for the pure Ni layer was ~0.2. However, the CoFs for Ni/SiC coatings were much higher: 1.22 for 5 g/L SiC and 1.38 for 10 g/L SiC. Subsequently, a decrease in CoF to 1.2 with an increase in SiC concentration (from 10 to 30 g/L) was found. The reason for the increment in the CoF with the increase in the particle content in the electrolyte from 5 to 10 g/L was not clear; it could be due to the insufficient load carrying effect of SiC particles [37]. N. P. Wasekar et al. prepared Ni–W/SiC composites with different SiC concentrations of electrolyte. Similarly, the CoF decreased by 1.5 times in the presence of SiC. In addition, the wear rate and CoF as a function of hardness were investigated. Both parameters showed derogation from the Archard law; in particular, the CoF increased with the increase in hardness. In this article, it was stated that the Archard wear law was not valid for Ni–W/SiC MMCs [58].

On the other hand, the reinforcement vol.% can be changed by additives in the electrolyte ([36,80,109]). N. K. Shrestha et al. tested the degree of the co-deposition of SiC particles with the Ni matrix in terms of the surfactant concentration in the electrolyte containing the azobenzene group (AZTAB). Pure Ni layer Ni/SiC composites (with different AZTAB concentrations in the electrolyte) were prepared. An increase in SiC vol.% in the composite was achieved with an increase in AZTAB in the electrolyte. To obtain CoF and wear behavior, a pure Ni layer and five Ni/SiC composites (26.2–71 vol.% SiC) were researched. A correlation between the concentration of SiC and the static CoF of the coating was not observed (CoFs from 0.205 to 0.225). The CoF of all the MMCs containing up to 62 vol.% were nearly identical to that of the Ni layer without any SiC reinforcement particles. However, the CoF of the 71-Ni/SiC MMC (value 0.225) was higher than that of other MMCs in this paper. This was probably due to the agglomeration of reinforcement particles in the deposits. This resulted in increasing the surface roughness of the MMC [80]. Similar investigations were conducted by F. Kılıc et al. Cetyltrimethylammonium bromide (CTAB: 0, 100, 200, 300, 400 mg/L) was used to obtain varying SiC vol.% in the Ni/SiC composites. It was observed that the SiC vol.% in Ni/SiC increased with an increase in CTAB in the electrolyte (1.26–11.37% vol.% SiC). The CoFs were approximately 5–7 times higher compared with the values obtained in a previous study (resulting CoFs: 1.0–1.4). However, there was no clear relationship between CoF and CTAB. The trend of plastic deformation in the roughness connections probably resulted in a higher and more unstable CoF [36]. On the other hand, K.H Hou et al. described the influence of SiC particles and the CTAB concentration in the electrolyte on the tribological properties of Ni/SiC composites. The average CoF decreased with an increase in SiC vol.% in MMC. As a consequence, the average CoF decreased (from 0.56 to 0.36) with an increase in the CTAB concentration in the electrolyte [80].

In addition, the reinforcement vol.% can also be changed by a change in the electrodeposition current parameters. P. Gyftou et al. (see Section 4.5.1) prepared Ni/SiC (DC) and Ni/SiC (PC) composites. They denoted a decrease in vol.% with an increase in the duty cycle. For DC power supply, the vol.% was stable [72]. H. Gül et al. (see Section 4.5.1) investigated the influence of DC [38,39] and PC [39] on the vol.% and friction properties. The influence of the electrophoresis/pulse electrodeposition of Ni–Co/Al_2_O_3_ vol.% and friction was described by Z. Jia et al. (see Section 4.5.1) [113].

The effect of vol.% or weight percentage (wt.%) was also investigated for Ni-alloy/MoS_2_ composites ([55,92,102,103,104,105,115]). Y. Zhang et al. researched Ni–Co/Al_2_O_3_–MoS_2_ composites with different levels of MoS_2_ concentrations in the electrolyte (see Section 4.5.1) [104]. Y. Wang et al. prepared Ni/Al_2_O_3_MoS_2_ MMCs with different MoS2 concentrations in the electrolyte: 0, 0.5, 1, 1.5, 2 g/L. In that paper, the authors described an increase in the wt.% of MoS_2_ in composites with an increase in the MoS_2_ concentration in the electrolyte (5.23–10.09 wt.%). A relatively low value of CoF was reached in the case of no MoS_2_ particles being used, which was explained by the relatively high MMC hardness value resulting from the high content of Al_2_O_3_ particles. A satisfying lubrication film was observed for the MoS_2_ content of 1.0 g/L. The usage of this concentration reduced the real contact area of the MMCs and a positive lubrication effect was observed for the lowest value of CoF. However, the CoF tended to increase when the MoS_2_ content was 1.5 g/L. The soft particles did not reinforce the MMC. In addition, wear products could increase the wear. The deformation energy increased the CoF value [115]. Similar results for CoF were found by M.F. Cardinal et al. for Ni–W/SiC composites [55] and by D. Trabelsi et al. for Ni/MoS_2_ MMCs [105].

The increase in the weight percentage of MoS_2_ in Ni/MoS_2_ composites (23–38 wt.%) with the increase in MoS_2_ in the bath (1–4 g/L) was also observed by S. M. J. S. Shourije et al. The CoF was decreased from 0.35 to 0.08 by increasing the MoS_2_ particles in the coating. In addition, it was shown that by increasing the MoS2 concentration in the MMC, two stages appeared in the friction tests: an initial low-friction stage (lubrication) and a state with a relatively high friction (lack of lubrication, perhaps the presence of MoO_3_) [103].

Y. He et al. focused on the mechanical and tribological properties of Ni–P/MoS_2_ composites. Four values of MoS_2_ concentration—3, 7, 10, and 20 g/L—were used to prepare the mentioned MMCs. The MoS_2_ weight percentage in the coatings was 2, 4.1, 7.9, and 7.1 wt.%, respectively. The CoF of the Ni–P coating was stable at 0.45; all the other composite coatings were stable much lower. In addition, the CoFs of the MMCs decreased by increasing the content of MoS_2_ in the coatings. The lowest CoF was observed for a 7.9 wt.% reinforcement in coating, equal to 0.05. In this work, it was explained that the concentration of MoS_2_ particles in the MMC has a significant impact on the surface roughness, which decreases (from 8.9 to 2.0 μm) with an increasing reinforcement weight percentage in the coatings. As a consequence, the CoF decreased with a decrease in the MMC surface roughness ([92,102]).

The effect of normal load on the CoF values is shown in Figure 4, and the influence of the reinforcement volume fraction on the CoF in Ni-alloy MMCs is illustrated in Figure 5.

#### 4.5.3. Effect of Reinforcement Particle Size (RPS)

The reinforcement particle size (RPS) has been determined in many papers. Micro- ([48,50,55,58,59,78,80,101,102,103,104,105,106,113]) and nano-size particles ([38,39,54,88,89,90,108,112,114,116]) were used to prepare nickel-alloy MMCs. Subsequently, the mechanical-tribological properties were investigated. In several papers, the MMC consisted of RPS in the range of, e.g., 100–1000 nm ([36,37]). However, the effect of micro- and nano-RPS on the CoF was investigated in a relatively narrow range ([41,49,72,75,107]).

I. Garcia et al. prepared Ni/SiC MMCs with 300 nm, 700 nm, and 5 μm RPS, respectively. For the two smallest RPS, they observed (after the running-in phase) CoFs equal to 0.29, and CoFs lower than 0.34 were obtained in an MMC with a 5 μm RPS at a comparable vol.% of co-electrodeposited particles. With the increasing amount and size of wear debris, the abrasive wear of MMCs increased. In addition, for each RPS investigated the CoF increased with an increase in the vol.% of SiC particles in the MMCs from 0.34 to 0.47 for 5 μm RPS and from 0.28 to 0.30 for 700 nm RPS [75]. Similar phenomena were described by P. Gyftou et al. (see Section 4.5.1). For both direct and pulse electrodeposition, the CoFs for Ni/SiC composites with 20 nm RPS were lower than the CoFs for MMCs with 1000 nm PS [72]. Two kinds of RPS (40 and 300 nm) were used by R. Li et al. to prepare Ni/SiC composites. They observed the influence of RPS in MMCs on the CoF; for 40 and 300 nm RPS, the CoFs were equal to 0.45 and 0.6, respectively. In comparison, the CoF for pure Ni was 0.85. R. Li et al. stated that the hard SiC particles in the MMCs reduced the contact between the MMC and the steel ball. As a consequence, the CoF values were lower. The presence of a third body (such as rolling friction with SiC particles) was also beneficial for lower CoF values [41].

M. Srivastava Sr. et al. prepared Ni/SiC and Ni-Co/SiC composites with two kinds of RPS: 25 nm and 1 μm. In addition, composites with different concentrations of cobalt and nickel were prepared. However, no significant modification in CoF values was found in the Ni–Co MMCs with reference to the SiC RPS and Co concentrations. On the other hand, the CoF for Ni/SiC with a 1 μm RPS was lower (0.707) than that for Ni/SiC with a 25 nm RPS (0.835). Perhaps this was due to the effect of the plastic Ni matrix, with its low particle concentration as compared to other MMCs [107].

D. Ahmadkhaniha et al. investigated Ni–P/SiC composites with two particle sizes (50, 100 nm). However, an influence of particle size on the CoF in MMCs was not observed [49].

The effect of reinforcement particle size (SiC, Al_2_O_3_, MoS_2_, Si_3_N_4_, Al_2_O_3_-MoS_2_) on the CoF in Ni-alloy MMCs is shown in Figure 6. The border between micro- and nano-particle size is 100 nm. Both SiC micro- and nano-particle sizes (black points) were used to prepare the aforementioned MMCs. The range of CoF for Ni-alloy/SiC is higher at the nanoscale (0.2–0.85) than at the microscale (0.15–0.6). It was observed that nano-sized Al_2_O_3_ particles (blue points) are mostly used to prepare Ni-alloy/Al_2_O_3_ MMCs, while micro-sized MoS_2_ particles (red points) are used to prepare Ni-alloy/MoS_2_ composites. A good way to reduce the CoF values in Ni-alloy MMCs is the usage of Al_2_O_3_–MoS_2_ reinforcement particles with a size in the range of 100–300 nm (blue/red color).

#### 4.5.4. Heat Treatment (HT)

One of the methods used to modify tribological properties in MMCs is heat treatment (HT) ([49,50,78,89,108,110]). S.-C. Wang et al. investigated the CoF for Ni-layer and Ni/SiC and Ni/Al_2_O_3_ composites. Firstly, six samples were prepared; subsequently, Ni layer and Ni/SiC and Ni/Al_2_O_3_ MMCs were heat-treated at 400 °C for 24 h. After this, the mechanical and tribological properties were investigated. The CoF was independent of the wear experiment time for HT at 400 °C. Comparing the Ni, Ni/SiC, and Ni/Al_2_O_3_ layers and their CoF values, pure Ni had the smallest value, but the CoFs values were similar. Slightly higher CoFs for Ni/SiC and Ni/Al_2_O_3_ MMCs were expected, due to the fact that the presence of RPS induced greater lateral forces [89].

Q. Wang et al. prepared as-plated Ni–P/SiC coatings (at 50 °C) that were subsequently annealed at four temperatures: 350, 400, 450, and 500° C (5 °C per minute). The mean steady CoF decreased with increasing temperature: values of 0.54, 0.55, 0.54, 0.52, and 0.51 were found for 50, 350, 400, 450, and 500 °C, respectively. In addition, the mechanism of friction with increasing time was analyzed. It was noticed that, for the highest temperatures, the running-in period was up to 3–4 times lower than that for composites plated at 50 and heat-treated at 350 °C. HT at high temperatures resulted in a smoother surface morphology, which consequently shortened the time needed to remove large asperities [108].

In recent years, D. Ahmadkhaniha et al. have focused on Ni–P coatings and Ni–P/SiC composites, including HT. The above samples for three temperatures (300, 360, 400 °C) were heat-treated. For as-plated samples, they observed stable but higher CoFs for the composite, in comparison with the Ni–P coating (relative 20%). Subsequently, HT samples were investigated. For a 300 °C HT, inverse phenomena in comparison with the as-plated samples were found. The composite CoF was relatively approximately 3.5% lower than Ni–P (300 °C). With higher temperatures, the relative CoFs were lower and stabilized at similar values. It was found that the friction mechanisms involved were mostly abrasion and tribo-oxidation and the formation of lubricant Fe oxides. Fe oxides combined with SiC particles mitigated this in the composite coating [50]. One year earlier, D. Ahmadkhaniha et al. investigated similar samples and the same HT conditions, but used different SiC particle sizes: 50 and 100 nm were used to prepare coatings and composites. The SiC particle size (50 or 100 nm) did not influence the CoF values. However, the CoF decreased with HT. The CoF was higher in the as-plated MMCs; in addition, it reached a stable value three times slower than in HT composites. It was found that the lowest CoF was attributed to the higher hardness of the HT MMC. This resulted from the precipitation of the stable Ni_3_P phase and the crystallization of the Ni matrix [49].

A. Martínez-Hernández et al. also investigated Ni–P/SiC composites. As-plated and heat-treated composites at temperatures of 300, 400, 500, and 600 °C (for 60 min) were prepared. At the end of the sliding measurement, the CoF values were 0.35, 0.5, 0.5, 0.35, and 0.4, respectively. Therefore, HT at 300 and 400 °C increased the CoF, in comparison to the as-plated CoF; however, HT at 500 °C caused a decrease in CoF to 0.35, equal to that of as-plated MMC [110].

N. P. Wasekar et al. prepared Ni–W alloys and Ni–W/SiC (5.92 vol.%) coatings. Subsequently, two samples were heat-treated at 500 °C in a vacuum. The CoFs analysis showed the influence of both parameters on the friction behavior. As-plated Ni–W/SiC MMC had a 27% lower CoF than the as-plated Ni–W layer. For samples heat-treated at 500 °C in a vacuum, the CoFs were lower compared with the as-plated samples (for Ni–W/SiC HT approximately 7.5% lower CoF than as-plated Ni–W/SiC). Perhaps the wear resistance of the composites increased after HT due to the improved hardness as compared to the as-plated layers. As a consequence, the CoFs were lower for the HT Ni–W/SiC composite [78].

### 4.6. Critical Overview

Friction is commonly measured; however, more attention should be paid to enable proper analysis. One of the parameters that can be significant for the tribological properties of MMCs is surface roughness. This parameter was measured in several papers ([49,54,58,78,90,102,105,108,109]). In addition, composites after deposition were polished ([75,88,89,90,106,107,108,117]); however, the surface roughness was reported only in papers ([88,108,109]). In the future, the influence of the surface roughness before and after polishing on the tribological properties of Ni-alloy MMCs should be reported. In this review, the importance of the interface is strongly underlined. The influence of the interface between the Ni-alloy matrix and the reinforcement on the mechanical and tribological properties was mentioned or reported in a narrow range in numerous papers ([38,39,49,54,59,78,89,104,106,114,118]). It would be beneficial to broaden the scope of measurements with direct interfacial strength measurements.

As shown in Figure 5, the range of reinforcement volume fraction was 0–20 vol.%. (see Section 4.5.2). The tribological study of Ni-alloy MMCs should be continued for higher reinforcement volume fractions, particularly those above 20 vol.% It should be noted that the reinforcement vol.% has an influence on the roughness, wear, and friction coefficient. As mentioned in Section 4.5.3, the CoF comparison between micron- and nano-sized particles was investigated in a narrow range ([41,49,72,75,107]).

The reviewed articles do not use statistical models to investigate the actual impact of the combined experimental factors on the CoF (no statistical tests were performed for Ni-alloy/SiC or Al_2_O_3_ or MoS_2_). Usually only typical CoF–normal load or CoF–particle vol.% graphs are presented. The results are interpreted in terms of the physical phenomena occurring in the friction pair. However, the combined effect of the tested parameters on friction could be supported by statistical tests, as mentioned in Section 4.5, where the test for the aluminum matrix was described.

The influence of different reinforcement particle sizes in Ni-alloy MMC on friction was not investigated. A nickel-alloy composite with different particle sizes could have interesting tribological properties (especially CoF). Three kinds of Ni-alloy MMCs with nano-, micro-, and micro + nano-size particles could be prepared and a comparison between their tribological properties could be made.

## 5. Interfacial Aspects

### 5.1. Interfacial Aspects in Co-Electrodeposition Particulate-Reinforced Metal Matrix Composites (CED PRMMCs)

The last section of this review considers the influence of the interface in CED PRMMCs on the overall properties of the material. It should be noted that in CED, there are four different interfaces: reinforcement-solution (RS), deposit-solution (DS), matrix-reinforcement (MR), and substrate-deposit (SD). In a given composite system, the MR interface quality has a crucial role in the behavior of the material. However, its quality results from the properties of RS and DS interfaces. RS and DS interfaces can be investigated to understand the processes occurring during the fabrication or they can be modified to amplify the features of material of particular interest. Both physical and mechanical properties can be tailored for interface modification. Parameters measured are wettability, zeta potential, chemical composition, and mechanical strength of the interface.

The wettability of reinforcement by solution plays a significant role in co-deposition mechanisms. Hydrophobic particles tend to leave a nano-metric space in between the particle and substrate which promotes co-deposition, whereas hydrophilic particles allow a solution to penetrate this space, which inhibits co-deposition [119]. The wettability properties depend on the investigated system, but typically can be changed with the chemical treatment of the reinforcement—oxidation, etching, deposition of transitional layer on the reinforcement.

Zeta potential is a typical electro-chemical indicator of particles, especially nano-sized, with a tendency for agglomeration. Combined with a high specific surface area of nano-particulate reinforcement, it is responsible for difficulties in obtaining repeatable dispersions of reinforcement in electrodeposition baths. Typically, a proper dispersion is believed to be achieved by the continuous stirring of the bath, ultrasound treatment before and during deposition, and the addition of surfactants [36].

Chemical reactions are yet another factor influencing interfacial aspects. The most common reaction would be oxidation on the surface of the reinforcement. It is worth mentioning that oxidation can take place before PRMMC preparation, during the CED, and it can also be performed as an additional step of PRMMC fabrication [120].

The mechanical strength of the MR interface has a direct influence on the mechanical and tribological properties of MMCs. However, it is rarely discussed for particulate reinforcement composites, and there are a couple of methods used in the literature. For the fiber–matrix interface, pushing individual fibers out of the matrix is a typical approach. It should be noted that careful sample preparation is crucial. To ensure reliability, numerous samples, in situ tests, and numerical analyses should be combined.

For CED PRMMCs, the MR interface strength is mainly constituted by adhesion force with little or no interphase, contrary to MMCs fabricated in elevated temperatures where chemical reactions between the matrix and reinforcement lead to significant amount of interphase [89,90].

A general review of the interface bonding strength in PR MMCs is given in [13]. It can be concluded that experimental approaches for PR MMCs cannot be easily transferred from fiber-reinforced MMCs and need further development, whereas models tend to possess some predictive features but need to overcome oversimplifications. Studies have been carried out with analytical models [121,122], the finite element method [68,123,124], statistical models [125], as well as molecular dynamics [126,127,128]. A detailed review of micromechanical models provides valuable insights and convinces us of the benefits of a statistical synthetic model [129].

### 5.2. Interface Measurements

Most work on interfaces for MMCs was carried out on Al matrices, and only a few papers have investigated in detail the interfacial phenomena in CED MMCs. However, methods used for PRMMCs fabricated by other routes [1] can also be applied for co-electrodeposited coatings. Interfacial strength can be measured in normal and tangential directions. Measurements can be conducted for matrix-reinforced interfaces extracted from as-deposited MMC (as in [130]) or for interfaces “artificially” prepared for measurement (as in [83], where a machined sintered SiC cube was used for the feasibility of the measurement).

One method for the direct measurement of adhesion force in PR MMCs is proposed in [130], where chemical etching is used to in situ reveal the matrix–reinforcement interface with a tensile testing machine. This allows us to measure normal stresses needed for debonding. The main limitations of this method are particle size, the accuracy of the measurement of the force (the smaller the particle, the smaller the real contact area, the smaller the bonding force to be measured) as well as accuracy of the contact area measurement.

Another approach for direct mechanical investigation of the interface was proposed in [83], where the tensile testing of the interface (normal loading) as well as pull-out testing (shear loading) were conducted for Ni–SiC composite. The main drawback of this approach is the difference in the size of the contact area in interfacial testing and the particle size used in MMCs. Micron and submicron particles definitely could exhibit different bonding strengths than a face of the size 1 mm^2^ or more, which was tested in this research.

In [131], micro-pillars with a metal matrix–ceramic reinforcement interface were investigated. In situ compression tests with careful sample preparation enable the measurement of shear strength for a SiC–Al system. An extension of this approach was lately shown in [132], where microstructure with different complexities were studied. A discussion on pillar preparation can be found in [133] and a general experiment in [134]; however, for MMCs, the proper characterization—primarily the angle between the interface plane and the direction of the applied compressing force—of the interface seems to be a crucial step for insightful results. The influence of the particle size was investigated in [135]. Similar research could be applied for CED PR MMCs.

In [136], a study of a particle removal mechanism and damage behavior in PRMMC was conducted with FEM, where the interface is modelled by the cohesive zone model. A brittle fracture is confirmed experimentally, however this work does not exceed experimental scope for interface and interfacial strength is assumed to be the same as in [131]. In [137], the method for the tensile tests of composites is shown for nano-laminates. This method could be applied for interfacial tensile strength measurement for PRMMCs instead of shear strength. In [138], SiC–Al interfacial behavior was studied with scratch tests as well as with numerical analysis.

Besides the direct interfacial strength measurements, interface quality is said to be good if scanning electron microscopy (SEM) images do not show significant porosity. Some works extend this qualitative approach to transmission electron microscopy (TEM) observations [139,140]. However, there is a need to proceed with quantitative results. Such results would provide a framework for the comparison of the interfacial strength between different fabrication routes and interface modifications.

One of the methods to modify the interface in PR MMCs is the addition of the protective coating on the reinforcement particles prior to MMC fabrication route. Such an approach was proven useful in [141], where the PVD deposition of different metals was tested for sintered MMCs.

It is worth noting that the local volume fraction (agglomerations and clusters of particles) tends to influence the overall behavior of the composite [139,140]. To acquire the homogeneity and volume fraction, proper image analysis is required. The protocol for such an approach is given in [142] as well as systematic point count was introduced as a standard test method for volume fraction [143]. Interfacial strength can be also quantitatively concluded also based on stress–strain curves from tensile tests, as shown in [144].

### 5.3. Critical Overview

Most papers focus on the properties of the studied MMC, and discussion of interface quality is usually limited to SEM observation; lack of voids and porosity is regarded as “good bonding”.

In [145], it was shown that during the CED of Ni–SiC composite, the oxidation on the surface of SiC takes place and surface modification techniques were proposed: controlled oxidation [145], carbonizing [146,147], electroless deposition [14], or PVD [59]. To fully understand the influence of surface modification on the overall performance, more attention should be paid to interfacial phenomena, including direct interfacial strength measurements.

Usually, the dispersion of the reinforcement is imaged with SEM micrographs and it is tentatively commented that dispersion is homogenous. However, insightful measurements are needed to confirm that, and one of these methods is X-ray microtomography (μCT) as shown in [148]. Additionally, it is known that agglomerations tend to weaken the composite; therefore, more focus should be given to advanced imaging techniques, both destructive (with Focused Ion Beam) or non-destructive (with μCT).

## 6. Future Outlook

Based on the reviewed papers, some ideas for future work on PRMMCs hardness can be provided. It is apparent that there is still space for publications focused mainly on hardness or indentation in general. New experiments could be designed that would allow the separation of the influence of different mechanisms, such as particle, dispersion and grain boundary strengthening, lattice distortion, grain size, and the volume and weight fraction of reinforcement.

It would be beneficial to conduct a detailed investigation of imprints, and not only “from the outside” with the standard use of OM and SEM. One ambitious idea would be to employ 3D X-ray tomography to image the volume of material under the intended indentation area, conduct indentation, and again image the material with tomography. This would give insight into the relative movement of particles in proximity to the indent. Similar results could be achieved by indentation on a cross-section of a layer in situ SEM.

There are still many possibilities to enhance the modelling and simulations of co-electrodeposited composites. The most notable molecular dynamics approach has the potential to grow in the coming years. As the computational power of computers increases, the simulation of larger volumes will become feasible. This will allow the simulation of a real-life system with larger particles and actual grains in the matrix. The FEM method could also be improved by adding elements able to simulate grain boundaries. Simulations and modelling are rarely compared directly and they could complement each other, as shown in the literature [56]. Modelling can be used to predict other parameters of materials. For example, stress–strain curves simulated with input data from indentation can be used to predict the tensile strength of composites.

The investigation of the mechanisms behind the increasing wear resistance for PRMMCs for micro- and nano-particles is of particular interest, especially the differences in those mechanisms. Some work has been carried out in this topic, but it is still far from being fully understood [81,84]. Such distinction would also give a foundation for PRMMCs reinforced with both micro- and nano-particles. The aim would be to combine the beneficial effects of both. Due to the anticipated significant changes in the microstructure of such PRMMCs, direct interfacial measurements could be carried out to study the influence of nanoparticles at the matrix–microparticle interface.

Future directions for wear investigations of PRMMCs involve extending the scope of the vol.% of samples in order to find the minimum wear rate in the function of vol.%. This would be most interesting for the strongest MR interfaces; therefore, direct interfacial strength measurements should be conducted for comparison purposes. Such a comparison could involve composites with untreated, protected, oxidized, and etched particles, as well as composites prepared with special routes (DC and PC, circulating electrolyte, magnetic assistance, or with organic additives and surfactants).

The future directions for the friction investigation of PRMMCs, besides the ideas outlined above, could include gradient coatings. The first way to achieve this is to provide a transition from a hard to a soft matrix. Such a transition could decrease the CoF according to Bowden–Tabor theory. This could be achieved by changing the matrix grain size during deposition, changing the electric parameters, or decreasing the relative amount of additives (saccharin is commonly used to decrease the grain size of electrodeposited (ED) nickel) through adding electrolyte without the additive during the process. The second way to achieve this is to change the vol.% of particles with the deposit thickness. This could lead to better deposit–substrate adhesion if the vol.% increases from 0 at the beginning of the process. This could be achieved by simply adding the particles during the process, but dispersion could be a main concern. To overcome this issue, a circulating electrolyte approach could be employed.

The investigations of the effect of SiC particle size on CoF were carried out widely (see Figure 6, black points). On the other hand, micron-sized Al_2_O_3_ particles and nano-sized MoS2 particles were not often used to prepare Ni-alloy MMCs. We also suggest focusing on the tribological properties of Ni-alloy/Al_2_O_3_-MoS2 composites because of the lack of studies on this composites and the relatively low CoFs (see Figure 4, Figure 5 and Figure 6, blue/red points) ([104,115]). In several papers, Ni-alloy/SiC MMC heat treatment and friction behavior have been reported ([49,50,78,89,108,110]). Similar investigations should be continued for Ni-alloy/Al_2_O_3_ and Ni-alloy/MoS2 composites. In addition, the effect of ambient temperature during wear and friction testing on MMC CoFs could be reported, similar to the paper [48].

The future directions for direct interfacial strength measurements include investigations of various surface modifications techniques (oxidation, protective layer, and carbonization) as well as various CED routes (ultrasonic agitation, magnetic-assisted ED, circulating electrolyte, and surfactants). Other parameters that could be taken into account are particle size, particle hydrophobicity/hydrophilicity, matrix grain orientation, and deposit thickness.

Another direction is the development of hybrid composites. Combining the beneficial influence of at least two kinds of reinforcement particles could lead to outstanding results. One example is the exploration of a Ni–SiC + MoS_2_ system where hard SiC particles could increase both the hardness and wear rate, whereas MoS_2_ would decrease the CoF [149]. Such an approach could lead to interesting findings regarding wear–friction–vol.% dependence. For hybrid composites, direct interfacial strength measurements are also of significant interest. They could facilitate the optimization of such composite structures.

As shown in this review, improving the properties of MMCs is our main focus; however, there are some minor aspects of investigations which could be discussed in future papers:Coating-to-substrate adhesion measurements should be added to a standard framework. Improving composite properties without proper adhesion could be meaningless.To accurately compare different systems, including direct interfacial strength, a similar vol.% and dispersion should be considered, as agglomerations could decrease the properties.

## 7. Conclusions

The thorough study of the tribo-mechanical parameters of Ni-based CED PRMMCs with an interfacial standpoint has made some key findings, as listed below.

The various CED fabrication details used cause many inconsistencies throughout papers. Despite the numerous investigations, the influence of a reinforcement on the composite properties is not clear—the presence of particles changes the microstructure of the matrix, therefore a simple comparison of electrodeposited Ni and co-electrodeposited Ni–SiC composite (with the same process parameters), is not enough for insightful study.

The use of a wide range of available particle sizes and testing methods causes difficulties in drawing general conclusions.

For wear rate, we introduced normalized wear rate as a ratio of the wear parameter reported in a given work to the wear parameter in that work measured for the lowest reinforcement vol.% (usually 0%). This showed that wear rates tend to have a minimum, however studies that use a wider range of reinforcement vol. % are needed. Additionally, it can be tentatively argued that the higher the vol.% for the minimum wear rate is, the stronger the matrix–reinforcement interface is.

For hardness tests, usually there is a deficiency in basic parameters—the size of indentation imprints and their spacing are crucial for a proper analysis of the results concerning the size of particles. The maximal load chosen should be small enough to exhibit an indentation depth of less than 10% of the coating thickness and big enough to enable averaging over a representative volume element. AFM or OM images of the indent could be also of significant value for results analysis. Additionally, 3D-tomography or cross-sectional indentation could lead to insightful observations where interfacial strength plays a crucial role in the material behavior.

For friction tests, sample preparation should be given more attention. The roughness of the CED composite would be high due to the loose attachment of the reinforcement at the surface layer. Without proper polishing prior to a friction test, reinforcement particles can detach, form a hard third body, and increase the friction. Additionally, the stronger the interface, the higher the load to be transferred via reinforcement and supposedly the lower the CoF.

Statistical models were proven to be of significant value and should be employed in investigations where several parameters are investigated.

Despite the establishment of direct interfacial strength measurements, there is still a need to broaden and deepen the scope of such measurements. Firstly, this would allow us to study the mechanisms of deformation. Secondly, this would allow us to compare different manufacturing routes and optimize the properties of composites.

## Figures and Tables

**Figure 1 materials-14-03181-f001:**
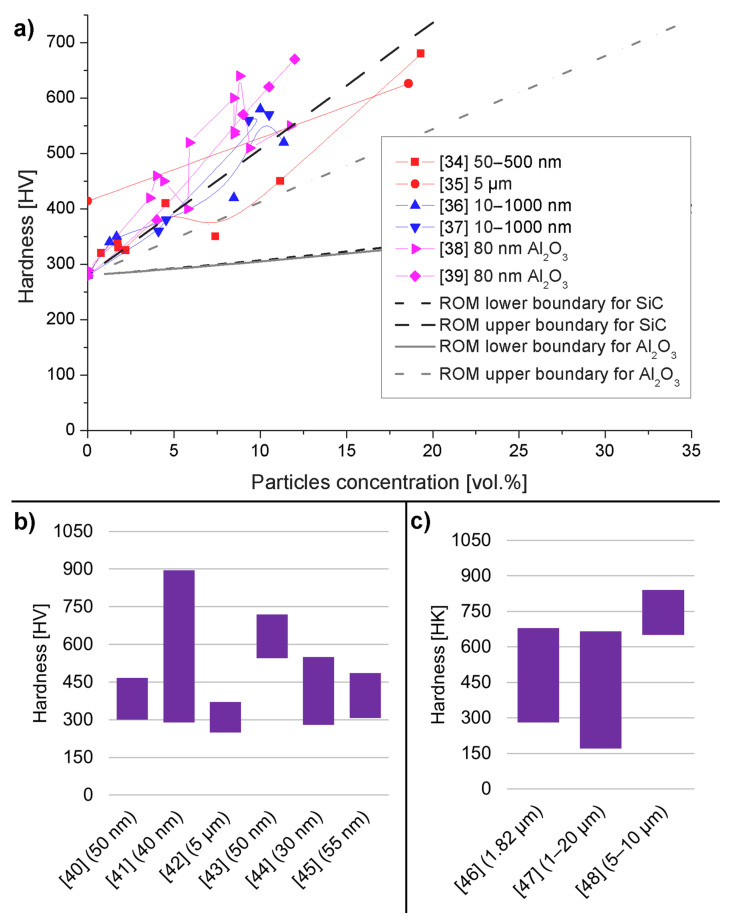
Results from the literature for pure nickel matrix with SiC reinforcement: (**a**) micro Vickers indentation with a specified volume fraction of reinforcement particles, (**b**) Vickers indentations without a specified volume fraction of reinforcement, and (**c**) Knoop indentation.

**Figure 2 materials-14-03181-f002:**
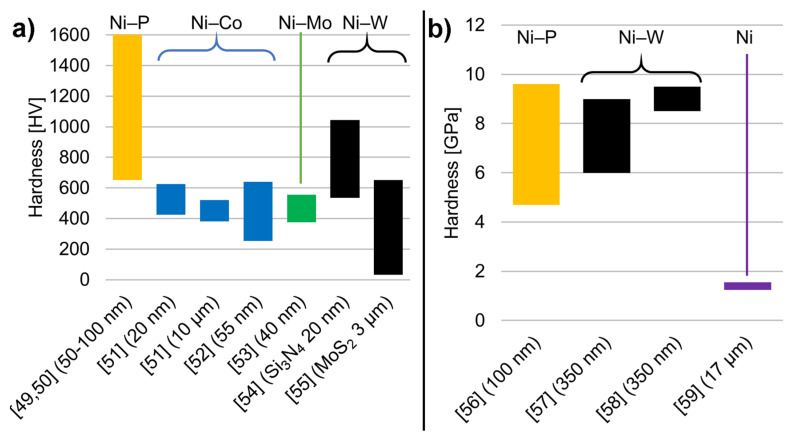
The hardness of composites with various nickel alloys as a matrix: (**a**) results of Vickers micro-hardness tests, (**b**) results of instrumented Berkovich nanoindentation with an exception for Jenczyk’s work, where instrumented Vickers microindentation was carried out.

**Figure 3 materials-14-03181-f003:**
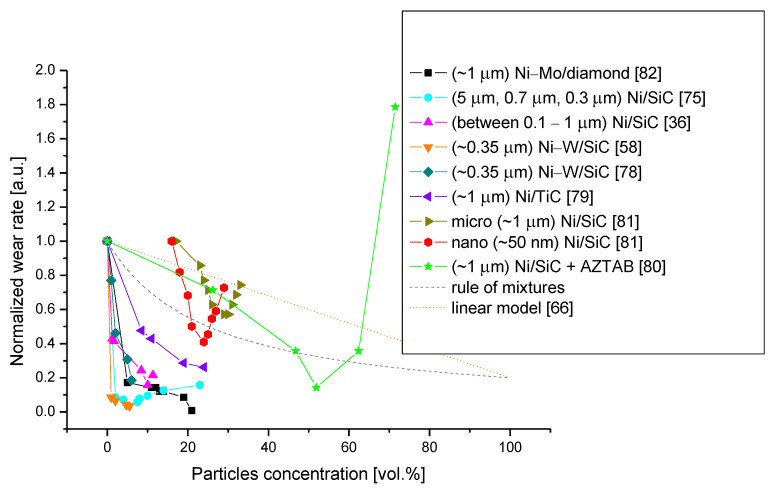
Normalized wear rate vs. particle concentration (vol.%) with indicated theoretical curves for the rule of mixtures and a linear model. The coefficients of friction (CoFs) graph includes friction measurements at room temperature for metal matrix composites (MMCs) without annealing.

**Figure 4 materials-14-03181-f004:**
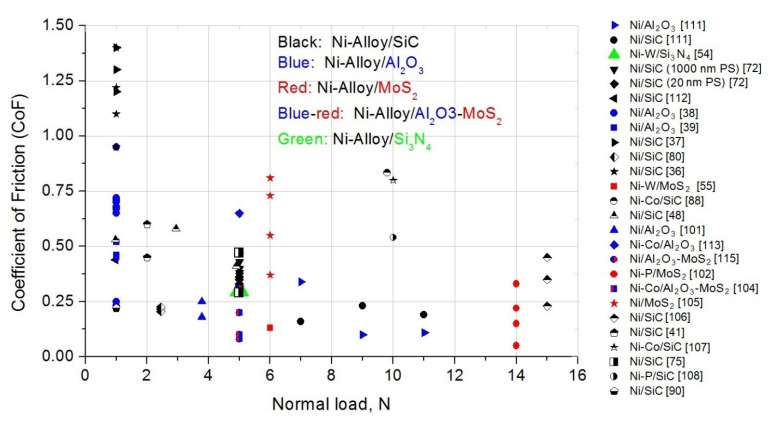
The effect of normal load on CoF in Ni-alloy MMCs. The CoF graph includes only dry friction measurements at room temperature for MMCs without annealing.

**Figure 5 materials-14-03181-f005:**
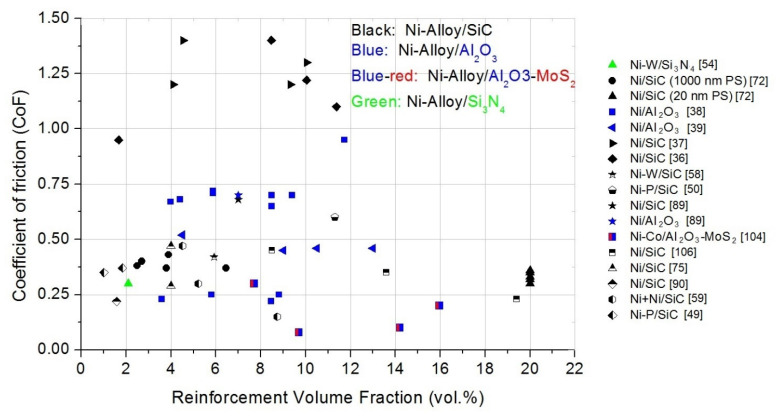
The effect of the reinforcement volume fraction on the CoF in Ni-alloy MMCs. The CoF graph includes only dry friction measurements at room temperature for MMCs without annealing.

**Figure 6 materials-14-03181-f006:**
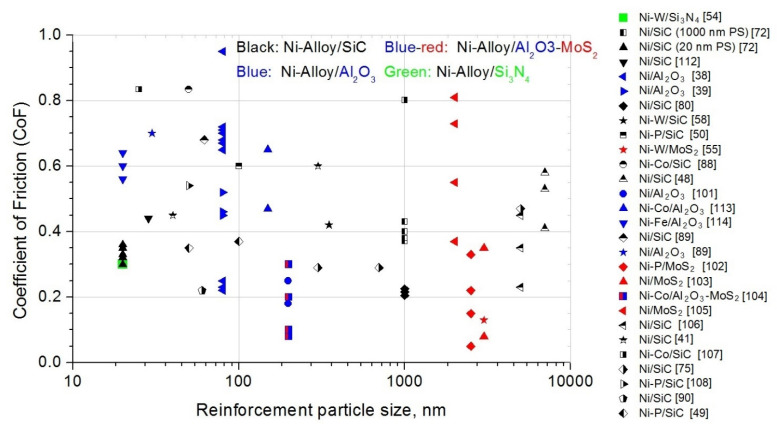
The effect of reinforcement particle size on the CoF in Ni-alloy MMCs. The CoF graph includes only dry friction measurements at room temperature for MMCs without annealing.

**Table 1 materials-14-03181-t001:** Summary of indentation hardness tests.

Procedure	Indenter Shape	Result and Unit	Measured Feature	Reference Standards
Vickers	4-sided pyramid	HV [kgf/mm^2^]	indent diagonals	ASTM E92—17 ISO 6507-1:2018
Brinell	sphere	BHN [kgf/mm^2^]	indent diameter	ASTM E10—18 ISO 6506-1:2014
Knoop	elongated 4-sided pyramid	HK [kgf/mm^2^]	longer indent diagonal	ASTM E92—17 ISO 4545-1:2017
Rockwell	sphere	HRA, HRB, HRC no unit	depth of indentation	ASTM E18—20 ISO 6508-1:2015
Vickers (micro)	4-sided pyramid	HV [kgf/mm^2^]	indent diagonals	ASTM E384—17 ISO 4516:2002
Knoop (micro)	elongated 4-sided pyramid	HK [kgf/mm^2^]	longer indent diagonal	ASTM E384—17 ISO 4516:2002
Instrumented (nano)	3-sided pyramid or sphere	[Pa]	indent area, force, displacement	ASTM E2546—15 ISO 14577-1:2015

## Data Availability

No new data were created or analyzed in this study. Data sharing is not applicable to this article.

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
