# Peer review of "Mechanical and Tribological Properties of Co-Electrodeposited Particulate-Reinforced Metal Matrix Composites: A Critical Review with Interfacial Aspects"

_materials, 2021, doi:10.3390/ma14123181_

Round 1
Reviewer 1 Report
The authors present a very nice review on the mechanical and tribological properties of electrodeposited PRMMCs with special attention to interfacial aspects. Overall, the review is very well written and would be very interesting for the readers. It includes detailed discussions on tribo-mechanical properties of Ni based PRMMCs. This critical review could be published in 'Materials'. Please see below my minor comments / suggestions.
- There are minor grammar corrections required for few sentences in the manuscript. (For example lines 16-17, 59 68, 302 etc.)
- Some of the sentences in the m/s are not well clear. It would be nice if they are rewritten clearly. (lines 362-363, 624-625, 1103-1105)
- lines 287-290, "No specific explanation for these high values is given apart from a general mention of particles strengthening, dispersal strengthening and grain refining, therefore this exceptional hardness should be attributed to the used plating solution". Any explanation how the exceptional hardness is correlated to the plating solution? what parameters of the plating solution (like composition, concentration, pH etc.) is responsible for this.
- It is noted that many of the figure numbers are wrongly quoted in the article text. It needs to be corrected. (For example, lines 371, 634 etc.)
- line 483, Where is this "figure 1"? Can't find it.
- A general comment, as a review focussing on co-electrodeposited composites, the readers would be interested to see details of the electrochemical co-deposition technique as a coating method, which seems to be missing in the review. However, it does not affect the quality/interests of the review.
Reviewer 2 Report
It is a really interesting study about the tribological properties of co-electrodeposited particulate reinforced metal matrix composites, congratulations.
However the reviewer is not a native English speaking person, the language seems OK.
General remarks:
- there are a lot of subscripts which are not in place, I tried to mark it to ca page 25, please correct it in the whole manuscript.
- summary diagrams are great but their usage could be significantly improved with a lot more informative captions, and font sizes should be unified.
- At the CoF and other tribological properties, the counterparts, load and lubrication circumstances etc. should be more pronounced, and on the summary diagrams should be emphasized which data can be compared based e.g. had the same counterpart, load etc.
I made my specific remarks, questions, comments in the manuscript_with_reviewers_comments
With proper corrections I think the manuscript can satisfy the publication criteria in Metarials.
